# Plausible Detection of Feasible Cave Networks Beneath Impact Melt Pits on the Moon Using the Grail Mission

Ik-Seon Hong and Yu Yi *

Department of Astronomy, Space Science and Geology, Chungnam National University, Daejeon 34134, Korea
* Correspondence: euyiyu@cnu.ac.kr

**Abstract:** In the future, when humans build their bases on terrestrial planets and their moons, caves will be the safest place for inhabitation. Large holes, believed to be cave entrances, have been discovered on the Moon, along with small features called "impact melt pits." In the Gravity Recovery and Interior Laboratory (GRAIL) gravity model, which is expressed in spherical harmonics (SH), it is difficult to express the gravity anomaly created by a small empty space below the surface. Nevertheless, we propose that a cave network, akin to an anthill, exists under the impact melt pits discovered on the Moon. This is because we think it is natural to apply a network created by Earth's small caves to the Moon. We obtained accurate Bouguer gravity measurements by calculating regional crustal density using localized admittance of the study area and detected weak gravity (mass deficit) information. By increasing the degrees and order of SH at regular intervals, we estimated the change in gravity at a specific position at high degrees and order, thereby extracting shallow depth information. To validate our method, we compared our results with those of existing studies that analyzed the previously known Marius Hills Hole (MHH) area. The analysis of seven regions in our study area revealed a mass deficit in some impact melt pits in four lunar regions (Copernicus, King, Stevinus, and Tycho). We propose that there is a cave network in this region, indicated by the gravitation reduction in the impact melt pits region. Our results can be useful for the selection of landing sites for future in situ explorations of lunar caves.

**Keywords:** lunar cave; lunar gravity; cave network

## 1. Introduction

When humans go into space, the Moon, in particular, a lunar cave, will be suitable as an outpost. There are typically three types of caves on Earth: solutional caves, sea caves, and lava tubes. Solutional and sea caves are created with water as the main medium, and lava tubes are created as a medium for lava. Recent studies have revealed the existence of water on Mars, and spectroscopic observations of the Moon have also revealed trace amounts of water on the lunar surface [1,2]. However, the presence of water on a large enough scale to be considered as a "water resource", such as those observed on Earth, have not been reported on both the Moon and Mars. As suggested by Lauro et al. [1], if there is liquid water on Mars, the existence of solutional and sea caves can also be considered. However, because there is no observational evidence of liquid water on the Moon, we can only consider the presence of lava tubes for investigating the existence of caves. The existence of lava tubes on the Moon has already been predicted from traces of lava activity on the lunar surface [3–5]. Lava tube regions have been proposed as suitable sites for human inhabitation on the lunar surface [6,7].

The role of a subsurface base is to defend against the physical damage that humans may experience on the lunar surface. Typical damages that we may experience on the lunar surface include diurnal temperature fluctuations, meteorite impacts, cosmic radiation, high energy particles, and dust. The daily temperature difference of the Moon varies according to latitude, with a change of about 100–400 K at the equator and a constant temperature of

about 240 K, even at the depth of ~0.4 m below the lunar surface [8]. The daily temperature difference on the Moon is very large compared to that of the Earth; thus, building a base below the surface may help in maintaining the temperature. Furthermore, a base built on the surface of the Moon may experience meteorite impact, resulting in base destruction. Moreover, meteorite impacts on the lunar surface can be observed from Earth using small telescopes [9]. The lunar crust, from the cave ceiling to the surface, can prevent damage from the primary impacts of meteorites and secondary impacts of eruptions [10]. Cosmic radiation is mainly an electromagnetic wave having high energy, e.g., ultraviolet, X-ray, and γ-ray, due to the influence of the Sun, and high energy particles include solar cosmic rays (SCRs) and galactic cosmic rays (GCRs). Both cosmic radiation and high energy particles cause damage to the human body; therefore, shielding is crucial for human habitation on the Moon. Angelis et al. [11] indicated that, on the Moon, the lunar surface is shielded from radiation only at a depth of 6 m and more. Thus, caves are the best place to completely block this radiation. Because lunar dust is fine grained and has low mass and the gravity on the Moon is low, it can float for a long time on the lunar surface. Therefore, dust may cause damage through skin contact or direct inflow into the lungs, and may even damage electronic equipment, especially circuits boards [12]. Lunar dust is the fine lunar soil floating on the surface of the Moon, as a result of electrostatic activity due to the solar radiation, and it can be explained by the dynamic fountain model [13]. Other causes for floating lunar dust are physical shocks, such as human steps or the movement of equipment. Thus, the insides of caves, which are undisturbed and where sunlight cannot reach, are ideal to avoid the generation of floating dust due to solar radiation.

Currently, in situ planetary cave exploration has not been performed, and exploration of planetary caves can only be carried out by remote sensing. In fact, it is the most commonly used technology to find the entrance of a cave. To find such an entrance, the main topographic observation object is a pit (or pit crater), which is a steep-walled negative relief feature [14]. A terrain presumed to be a planetary cave was discovered earlier on Mars before the observation of such a terrain on the Moon. Cushing et al. [15] used the images captured by the Mars Odyssey's Thermal Emission Imaging System (THEMIS) and found an area having less temperature variations than the surrounding area at night, revealing the existence of a void in this region. Since then, many pits have been found in several areas on Mars [16,17]. Haruyama et al. [18] discovered the lava tube skylight called the Marius Hills Hole (MHH) from the Terrain Camera (TC) of the Seleneological and Engineering Explorer (SELENE) launched in 2007 and observed a cave entrance. Observations from the Lunar Reconnaissance Orbiter Camera (LROC) Narrow Angle Camera (NAC) of the Lunar Reconnaissance Orbiter (LRO) launched in 2009 revealed two additional large pits, the Mare Ingenii Hole (MIH) and Mare Tranquillitatis Hole (MTH), and was found to have empty spaces inside, including the MHH [14,19,20]. Wagner and Robinson [14] found pits located in the highlands; they found a total of 221 impact melt pits having small sizes, irregular shapes, and clustering properties in the impact melt deposits derived from 29 craters near and inside the crater. Recently, studies have found a total of 257 pits from 34 craters; these data are currently being distributed as catalogs [21].

Lunar cave detection refers to the sensing of empty hollow space under the lunar surface; several previous studies have attempted using gravity exploration to deduce the internal structure of these caves. The Gravity Recovery and Interior Laboratory (GRAIL), launched in 2011, provided high resolution gravity field information about the Moon [22,23]. For the stable existence of the lava tube below the lunar surface, it must be a huge structure that is several kilometers wide, with a ceiling that is several hundred meters thick [24,25]. Chappaz et al. [26] found subsurface mass deficit using two types of gravity field data, free air anomaly (FA), and Bouguer anomaly (BA), and suggested the possibility of lava tubes in 11 sinuous rille areas, including the MHH.

Another method of subsurface exploration is the use of ground-penetrating radar (GPR). Kaku et al. [27] analyzed the MHH region's echo signal using the Lunar Radar Sounder (LRS) data, which are one of the payloads of SELENE, to discover the void under

the surface with a void height of 75 m roughly. However, Kobayashi et al. [28] denied the results of Kaku et al. [27], with respect to presenting the conditions and data processing methods required when searching for lava tubes using SELENE LRS data; to understand the use of this technology and its use in the detection of cave openings and its internal structure, the results of lunar cave exploration using GPR need to be investigated further.

In this study, we explored a lunar cave via a geophysical approach using the GRAIL gravity exploration data. However, we only targeted impact melt pits on the Moon, and not mares and highland pits, which have been the most common topics of several previous studies, because we think that it is natural to apply the network created by small caves on Earth rather than large single individual caves to the moon.

## 2. Data and Methods

### 2.1. Data

The GRAIL's gravity observation data are distributed as a gravity model consisting of spherical harmonics (SH) coefficients. From the early model, GRAIL420C1A, to the current model, GL1500E, gravity models of various degrees and orders have been developed over the past years [23,29–34]. In the gravity model, as a characteristic of SH, the higher the degree and order, the shorter the wavelength; therefore, shallow depth information can be effectively expressed using this model [35]. Additionally, when a gravity model is expressed as a map, it has a high spatial resolution, with high degree and order; therefore, such models can be used to deduce the gravity of a small area. The GL1500E model offers the highest degree and order among the currently published gravity models and is suitable for exploring lunar caves. However, Goossens et al. [36] improved the GRGM1200B gravity model (with a degree and order of 1200) and developed the GRGM1200B RM1 model, which provided a high degree and order; notably, GRGM1200B RM1 showed a significantly higher correlation between the observed gravity and the gravity derived from topography, compared to the GL1500E model. Thus, GRGM1200B RM1 expresses shallow depth gravity information more effectively, even though GL1500E has a high spatial resolution. Therefore, in this study, we proceeded with the GRGM1200B RM1 for $\lambda = 1$ model, which provides more detailed shallow depth information, even if it compromised the spatial resolution to some extent.

### 2.2. Method

#### 2.2.1. Hypothesis

Before analyzing the gravity model, it is necessary to know how the existence of a small lunar cave changes the gravity field. In this study, we determined whether the information of a small subsurface void can be expressed in a gravity model. Lowrie [37] describes the change in gravity in a simple form, with respect to the differences in the density, radius, and depth of a mass, when an object having a different density from the perceptual density exists below the surface. The equation of Lowrie [37] was used, modified for our study:

$$\Delta g = \frac{4}{3}\pi G \Delta \rho R^3 \left(\frac{1}{x^2 + y^2 + z^2}\right),\tag{1}$$

where $R$ is the radius of the void, $x$, $y$ is the distance from the center (surface above the void position), $z$ is the depth, $\Delta \rho$ is the density variation, and $G$ is the gravitational constant.

Based on this, the changes in gravity, with respect to the density of the lunar crust (2550 kg/m$^3$), density of empty space (0 kg/m$^3$), radius (1–13 m), and depth, are shown in Figure 1a. When the radius of the empty space was more than 5 m, the gravity change appeared at a depth of several tens of meters, and this amount of change was expressed in the gravity model effectively. Next, we checked the spatial change in gravity that appeared under specific conditions. Figure 1b portrays a gravity anomaly at void center $g_0$ when there was a spherical empty space having a radius of 5 m at a depth of 20 m, with the lunar crust as a background. As the distance from the empty space increased, the reduced gravity recovered, and the shape of the wavelength showed a sharp pin, rather than a gentle curve.

This wavelength corresponded to SH; notably, the degree and order must be approximately 100,000 or more to be expressed effectively. A small lunar cave corresponding to this degree and order cannot be expressed in the currently used gravity model. Therefore, we hypothesized that if a small lunar cave exists, it will form a network (akin to an anthill), with multiple entrances, rather than resembling a single entity, such as the MHH, MIH, and MTH. If subsurface voids are entangled similar to a net in a certain area, such as the lava tubes observed on Earth, the change in gravity in the area having empty spaces can be sufficiently detected in current the gravity field model. In addition, because the lunar crustal structure is very complex, the small signal of the lunar cave will most likely be expressed in a combination of various high degrees and orders.

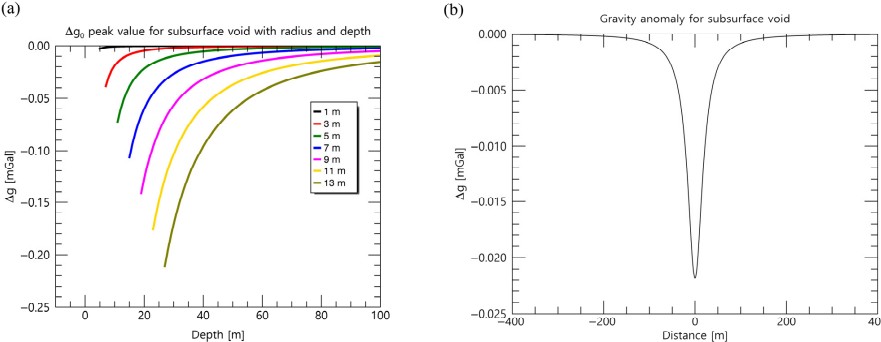

**Figure 1.** (**a**) Variation of gravity anomaly depending on radius and depth of lunar subsurface void. $\Delta g_0$ is the surface above the void position where the change in gravity is strongest, and the radius is indicated in the legend by specifying the color from 1 m to 13 m (at intervals of 2 m). Lunar crustal density was set as the background density ($\rho_0 = 2550$ kg/m$^3$), and the density of the empty space was set as $\rho_0 = 0$ kg/m$^3$. (**b**) Gravity anomaly when an empty sphere having radius of 5 m exists at depth of 20 m. Gravity anomaly variation decreases sharply as the distance between the empty space and the surface increases and decreases by 95 % when compared to the $g_0$ point at 100-m depth. Density conditions of (**b**) are same as those of (**a**).

### 2.2.2. Research Target

Our study area is an impact melt pit consisting of a terrain that has a high possibility of a subsurface lunar cave network. Unlike the pits in the mare and highland areas, this terrain is small, has an irregular shape, consists of several clusters that form near fractures, and is distributed in impact melt deposits [14]. In addition, because it has a similar shape to lava tube ceilings observed on Earth, it may connect to subsurface voids [15,38,39]. If there is a cave network beneath the impact melt pits, they can affect the gravity of the impact melt pits area, so we set the analysis target as an impact melt pit. Wagner and Robinson [14] found the presence of impact melt pits in a total of 29 craters on the Moon. We performed the analysis on seven craters, containing more than 10 impact melt pits (Table 1). In the latest lunar impact pit catalog released by Wagner and Robinson [21], there was a change in the number of pits by region. In the case of the Lalande crater, there were eight pits, not ten; however, our targets were based on the results of [14].

**Table 1.** Seven craters containing more than ten pits [14].

| Name | Latitude | Longitude | No. Pits > 5 m Wide |
|---|---|---|---|
| King | 6.5°N | 119.8°E | 51 (62) |
| Tycho | 43.3°S | 11.2°W | 31 (24) |
| Copernicus | 9.6°N | 20.1°W | 17 (32) |
| Lalande | 4.4°S | 8.6°W | 19 (8) |
| Stevinus | 32.5°S | 54.1°E | 16 (26) |
| Aristarchus | 23.7°N | 47.5°W | 13 (14) |
| Crookes | 10.4°S | 165.1°W | 11 (14) |

Note. ( ) in 'No. pits > 5 m wide' means a new reporting number from [21].

### 2.2.3. Estimation of Localized Crustal Density

In this study, we used SHTOOLS (https://shtools.oca.eu/shtools/public/index.html (accessed on 15 October 2021)), which is a software developed for a potential field operation (represented by SH) for gravity model analysis [40]. Previous representative studies of the lunar gravity field have mainly focused on subsurface structure and evolution, such as crustal thickness determination and dike exploration for the entire Moon [41,42]. In the case of gravity field analysis for the entire Moon, we set one value of crustal density when calculating the Bouguer gravity, which expressed only the gravity beneath the surface (by removing the influence of topography). However, due to the difference in the composition of the mare and highland crust of the Moon, the actual density varied (depending on the location). This difference affects the Bouguer gravity of a specific area, resulting in inaccurate analysis. Therefore, to analyze the localized Bouguer gravity, it is necessary to estimate the localized crustal density of the corresponding area.

In this study, we calculated the admittance to estimate the perceptual density at the seven craters. Admittance refers to the gravity anomaly generated by a change in topography [43]. Watts [43] explained the relationship between crustal density and admittance using four isostasy models (uncompensated, Airy, Pratt, and flexure). In the form of admittance, as the wavenumber (replaced by the degree and order of spherical harmonics, $k = (l + 1/2)/R$; $k$ is wavenumber, $l$ is spherical harmonics degree, and $R$ is radius) increased, the three models converged to the uncompensated model. Our target is a small terrain, and we needed to consider high degrees and orders. Therefore, in our study, we applied the uncompensated model, which is the simplest form and does not require assumptions about parameters to fit the Moon. The equation of Watts [43] was used:

$$Z(k)_{uncompensated} = 2\pi G(\rho_c - \rho_w)e^{-kd}, \tag{2}$$

where $\rho_c$ is oceanic crustal density, $\rho_w$ is density of water, $d$ is depth, $k$ is wavenumber, and $G$ is gravitational constant. However, this form of equation applies only to Earth. We need to modify this equation for Moon as follows:

$$Z(k)_{Moon} = 2\pi G\rho_c e^{-kd}, \tag{3}$$

where $\rho_c$ is the lunar crustal density.

When admittance is expressed as an equation, it is the value obtained by dividing the cross-power spectrum of gravity and topography by the power spectrum of topography. The equation of admittance spectrum is as follows:

$$Z(l) = \frac{S_{hg}(l)}{S_{hh}(l)}, \tag{4}$$

where $l$ is the spherical harmonics degree, $S_{hg}(l)$ is the cross-power spectrum of gravity and topography, and $S_{hh}(l)$ is the power spectrum of topography.

To determine the localized admittance, we computed a localized multitaper power spectrum for gravity and topography [44–46]. The MoonTopo2600p SH model was used to express the topographic data [47]. We set a spherical cap of a radius of 10° for the seven regions to calculate a localized multitaper power spectrum. The localizing window bandwidth was set at 100, and we used 44 windows, with a concentration factor > 0.99. Figure 2 portrays the localized admittance for the seven regions. By combining Equations (3) and (4), we estimate the local crustal density and depth through least square fitting. Table 2 portrays the crustal densities calculated for the seven regions for the range of degree and order being 250–600, with only a small difference between the isostasy models in each adjustment.

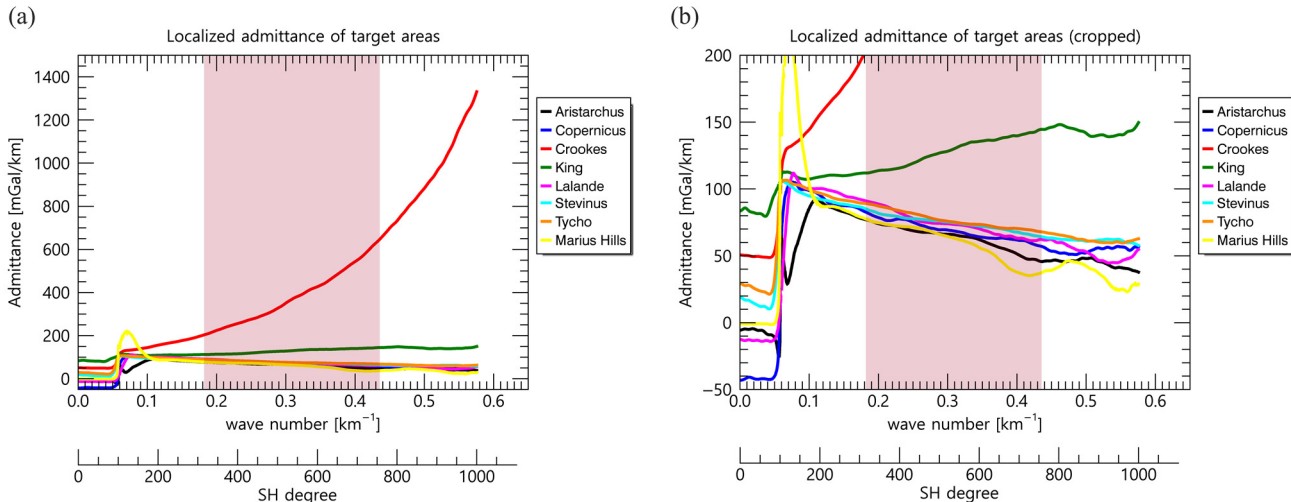

**Figure 2.** Localized admittance for each target region. Here, (**a**) represents all regions and (**b**) is cropped and expressed separately because the admittance of Crookes is extremely large to estimate the change in admittance in other regions. Crookes and King are located in the highlands (high elevation), and all other areas are located at mare or near mare. The admittances of Crookes and King increased due to the difference in topography (according to location). Marius Hills (yellow) was not a target area but an area for validating our method. The red color in the background was the range used to calculate the localized crustal density.

**Table 2.** Localized crustal density and depth of target areas.

| Name | Crustal Density [kg/m$^3$] | Depth [km] |
| --- | --- | --- |
| King | 2147.61 | −0.94 |
| Tycho | 2659.97 | 1.00 |
| Copernicus | 2618.09 | 1.23 |
| Lalande | 2933.59 | 1.35 |
| Stevinus | 2432.88 | 0.82 |
| Aristarchus | 2369.62 | 1.10 |
| Crookes | 2107.24 | −3.63 |

Note. In depth, negative value means higher elevation rather than lunar radius.

### 2.2.4. New Processing Method of the Gravity Model

In this study, we used a gravity gradient tensor analysis method suitable for short-wavelength identification, using the Bouguer gravity derived from the crustal density of each region [41,48]. In addition, it is necessary to apply degree and order filtering to the gravity model to detect gravity changes at shallow depths. We used a high-pass filter, which remove the degree and order values of 1–60 to exclude the regional trend [41]. The gravity was calculated by increasing the range by a regular interval, from the minimum degree and order. Because the filtering range increased, the calculated results incorporated the shallow depth information. If each result was combined, such as a structure of spectral image (e.g., visible or infrared), the changes in gravity can be observed according to the depth of any region. We calculate the horizontal eigenvalues for effective analysis of residual trends [41]. The eigenvalue of gravity gradient tensor indicated negative features for high density and positive features for low density. Calculating eigenvalues in this way indicated which feature change trend appeared in which degree and order, and the gravity changes at shallow depths were also identified efficiently. Figure 3 shows a simple conceptual diagram of this. We calculated the maximum degree and order starting from 65, increasing in intervals of 5.

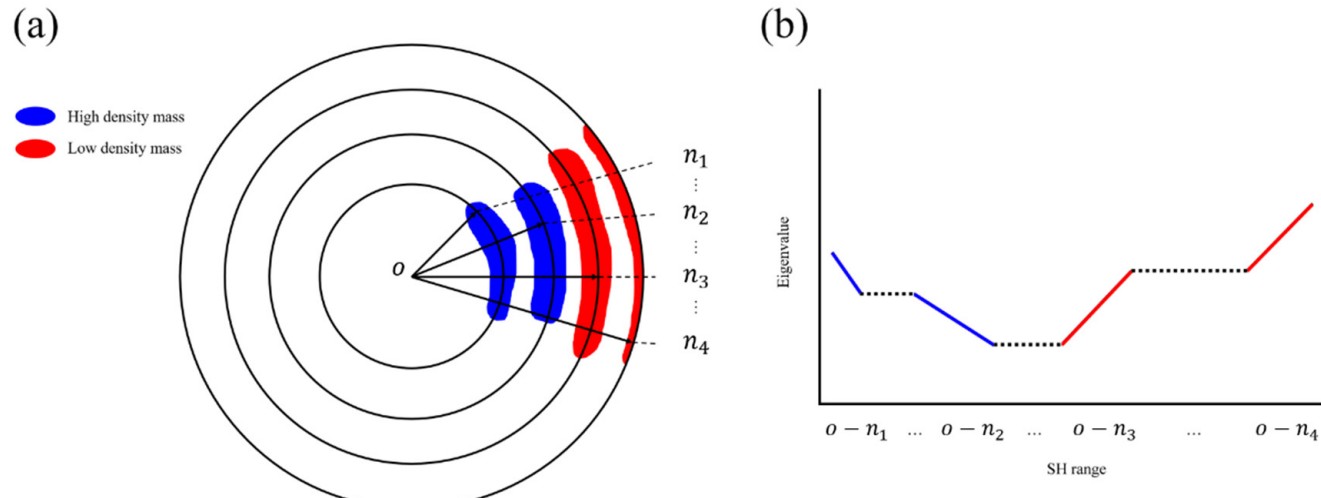

**Figure 3.** Concept of the method devised in our study. In the gravity model of spherical harmonics (SH), as the degree and order increase, a shallow short wavelength is expressed. Calculations by specifying a degree and order range can indicate the gravity at a specific depth level. It is easy to find out what information is added at a shallower depth by increasing the degree and order range at regular intervals. In arbitrary SH degrees $n_{1-4}$, if a mass exists in a specific degree and order (correspond to depth), with a structure similar to that shown in (**a**), it can be expressed as (**b**) through eigenvalues of gravity gradient tensor. In the range of degree and order corresponding to high density mass, eigenvalues indicated a negative trend, and in the low-density mass section, eigenvalues indicated a positive trend.

In this study, we used gravity model GRGM1200B RM1 for $\lambda = 1$ (degree and order 1200). However, it is not possible to calculate the gravity field up to degree and order 1200 blindly to obtain shallow depth information. Because the gravity model expresses a shorter wavelength with increasing degree and order, even though shallow depth information can be known, the effect of errors may increase. In this study, we determined the maximum degree and order. In previous studies, the power spectrum of Bouguer gravity revealed that as the degree and order increase, the power first decreases and then increases, which is a phenomenon that occurs when the error is larger than the main signal [23,29–34]. If there is a cave network beneath the impact melt pits on the Moon, interpretation at a high degree and order is required. Therefore, to exclude this error, we calculated the power spectrum of the localized Bouguer gravity for seven regions, and the degree and order of the point where the power increased was set to the maximum value to be used for filtering; the values are shown in Table 3.

**Table 3.** Maximum degree and order to minimize the error.

| Name | Max. Degree and Order |
|---|---|
| King | 606 |
| Tycho | 609 |
| Copernicus | 564 |
| Lalande | 567 |
| Stevinus | 603 |
| Aristarchus | 534 |
| Crookes | 612 |

### 2.2.5. Validation

We verified the proposed method by analyzing the MHH region, which has been significantly studied by Chappaz et al. [26]. The crustal density of the spherical cap centered on the MHH determined from the admittance at an angular radius of $10°$ is 2569.72 kg/m$^3$.

To ignore the error that appears as the degree and order increase in SH, the gradient of the power of the localized Bouguer gravity was calculated, and the maximum degree and order applied to the analysis are 507, with a gradient exceeding 0. The eigenvalue map for the 60 to 510 section including the maximum order 507 is shown, and the positive feature can be seen along with the rille form including the pit (Figure 4a,b) is the eigenvalue variation for the two pit regions, and the range of 50 is set based on 510 including the maximum order 507, and the red dot is MHH. The reason for placing the maximum degree and order over the range of 50 was to identify the positive trend that indicated the mass deficit at the maximum degree and order, which portrayed the shallow depth information, excluding errors, within a narrow range.

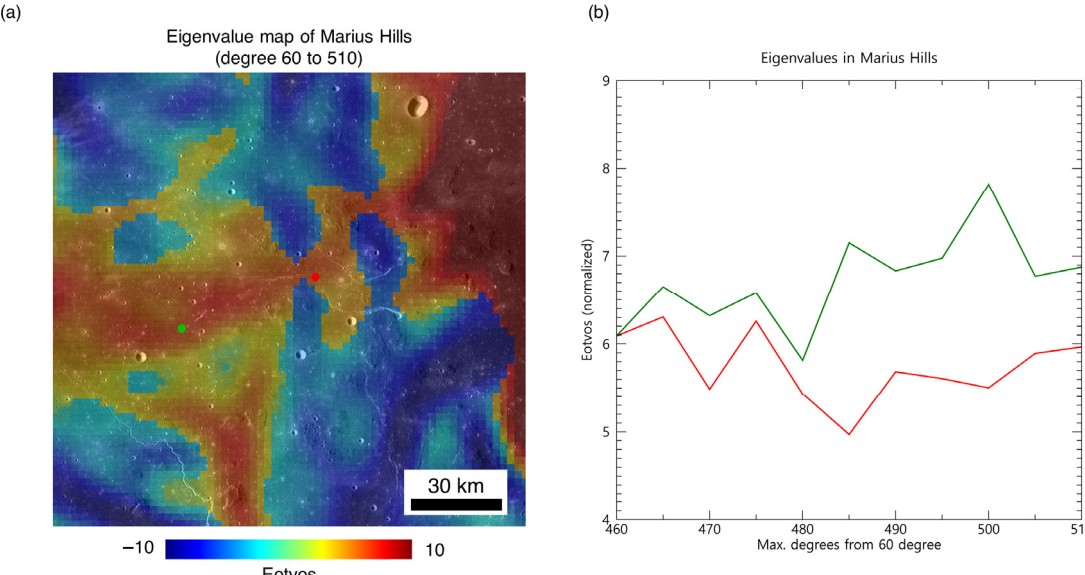

**Figure 4.** Eigenvalues map of Marius Hills Hole (MHH) region and eigenvalue variation of two pit locations. (**a**) A mass deficit (red color) appears in the rille area of the two pits. (**b**) As the eigenvalue variation for two pit points, the color of each plot corresponds to the color of the dot in (**a**). The increasing trend of MHH (red) appears after 483 SH degrees, and the positive trend is not clear compared to the green point. The graph is normalized at 460 degrees to make it easier to see the trend change. We identified the terrain using a Lunar Reconnaissance Orbiter Camera (LROC) Wide Angle Camera (WAC) image in (**a**).

The eigenvalue variation for the two pit points shows positive trends as the degree and order increase. Still, in the case of MHH (red dot), within the specified range, the trend does not increase but increases after 483 SH degrees. On the contrary, the increasing trend of the green point is more pronounced. From this result, it seems that the underground mass deficit of the green point is more apparent than that of MHH. Since it is not a cluster region of pits corresponding to our hypothesis, the cave network, there is a limit to the eigenvalue trend of MHH, strongly supporting our hypothesis.

Considering common sense, a mass deficit can naturally appear even in an area where there is no pit, so the verification result may be weak. However, we note that the cave network may contribute to several causes of mass deficit in the pit area. It is not possible to accurately distinguish caves from current data. Thus, in the best way, we verified the existence of the cave network in the pit area, which is one clue that can be accessed in the current situation.

## 3. Results and Discussion

In the existing literature, no study has explored small-scale voids, such as cave networks in terrestrial subsurfaces, by analyzing gravity field data. Therefore, we derived the results by presenting a new method for analysis. Fortunately, this method has been verified

to produce the same results, compared to the results of Chappaz et al. [26]. The magnitude of change in eigenvalues in the analysis results of the study area was small, and any errors may interfere with our interpretation. Therefore, we found the maximum degree and order, wherein the error would not be stronger than the gravity, through power spectrum analysis, and analyzed the change in gravity caused by the mass in the lunar subsurface, and thus, excluding the error as much as possible.

Then, we analyzed the variations in the eigenvalues over a specific range from the maximum degree and order for the seven regions. When interpreting the results, our main focus was to identify the variation trend of the eigenvalues, according to the degree and order, and not the eigenvalues distribution in the impact melt pit region. The color of the eigenvalues map in each region applied different stretches for improving the detection of eigenvalue variations and displayed the range of color bars.

As a result of the analysis, there are only four areas (Copernicus, King, Stevinus, and Tycho) where the increasing trend of the eigenvalue appears. The number of pits with increasing eigenvalue trends in each region is five in Copernicus, seventeen in King, four in Stevinus, and only one in Tycho. In the eigenvalue map, due to spatial resolution (2.2 km/pixel), a pit cluster is included in one pixel, so Copernicus and King are represented in three places, and Stevinus and Tycho are equal to the number of pits. In addition, the eigenvalues trend in the other pits area is not consistent with our hypothesis or it is not easy to interpret clearly in both the primary model and the clone model. In that example, the eigenvalues trend of all the pits in the King area (Figure A1) where the pits are most concentrated is shown in Appendix A.

For the GRGM1200B RM1 for $\lambda = 1$ model, there are a total of 100 clone models. Although we applied to filter to exclude high-order errors from the spherical harmonic function, we randomly selected five replicated models for more detailed error analysis and applied the same research method to check whether the same results are obtained. As a result of the clone model analysis, an increasing pattern of eigenvalues appears, as in the result of the main model, and it is confirmed that the error of the gravity model does not contribute to the SH range of our study. This shows that our results are not affected by errors and are reliable.

The clone model results (Figure A2) are shown in the Appendix A. Additionally, the clone analysis results of the Marius Hills region (Figure A3) are also attached to the Appendix A.

Copernicus is a complex crater on the near side of the Moon and is 96 km in diameter. The crater is located on the wall of the buried crater that has a different topography compared to the surface [49–51]. In Copernicus, most of the pits are clustered in the northern part of the crater plain, so a mass deficit was expected. Contrary to expectations, however, the analysis showed mass deficits in three regions that were not related to the main cluster (Figure 5a,b). All of these pits are in the red region corresponding to the buried crater wall, and in particular, four pits are gathered in the area with green and blue points, suggesting the possibility of a cave network.

Considering the presence of olivine in the central peak of Copernicus [52], we were able to deduce that the mantle material was ejected through both the mare and highland crusts. If so, the mantle should also be uplifted, and the eigenvalues of the region should show a negative trend. However, in Figure 5a, Copernicus has a positive feature (red color) for most of it. It is reasonable to view this as the influence of buried craters. The buried crater center has a crustal thickness of 15 to 20 km and the Copernicus center has a crustal thickness of 30 to 40 km [42]. Regarding crustal thickness, we can say there is definitely mantle uplift in buried craters, but not in Copernicus. However, since the olivine of the Copernicus central peak exists, the mantle uplift can be expected, but the trace seems to be hidden due to the buried crater wall.

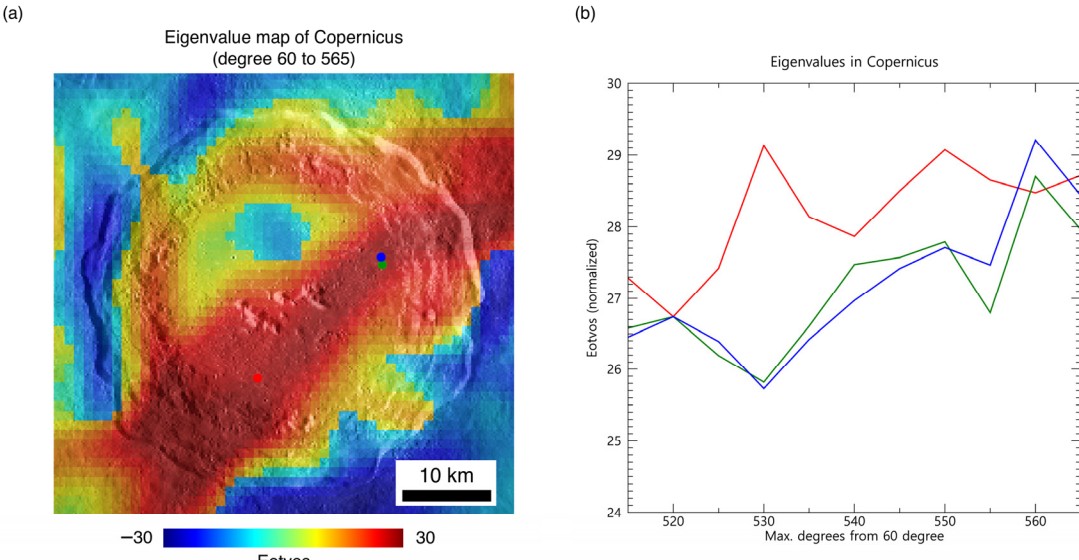

**Figure 5.** Eigenvalues map and eigenvalue variation of the Copernicus pits region. (**a**) In the positive feature corresponding to the wall of the buried crater, a mass deficit appeared in only three points, and the two points (green and blue) are adjacent to each other. (**b**) Green and blue points adjacent to each other show similar eigenvalue variation. The graph is normalized at 520 degrees to make it easier to see the trend change.

The King crater is located on the far side and is an impact melt deposit region; this includes a satellite crater called King Y located north of the main crater, with a diameter of 20 km. The impact melt pits in this crater are mainly distributed in the central and eastern parts of the impact melt deposit. There is a total of 62 pits in King, and a positive trend appears only in three regions (Figure 6a,b). Three (red), ten (green), and four (blue) pits are clustered in each area. Among them, it can be said that there is a high possibility of the existence of a cave network in the green point area.

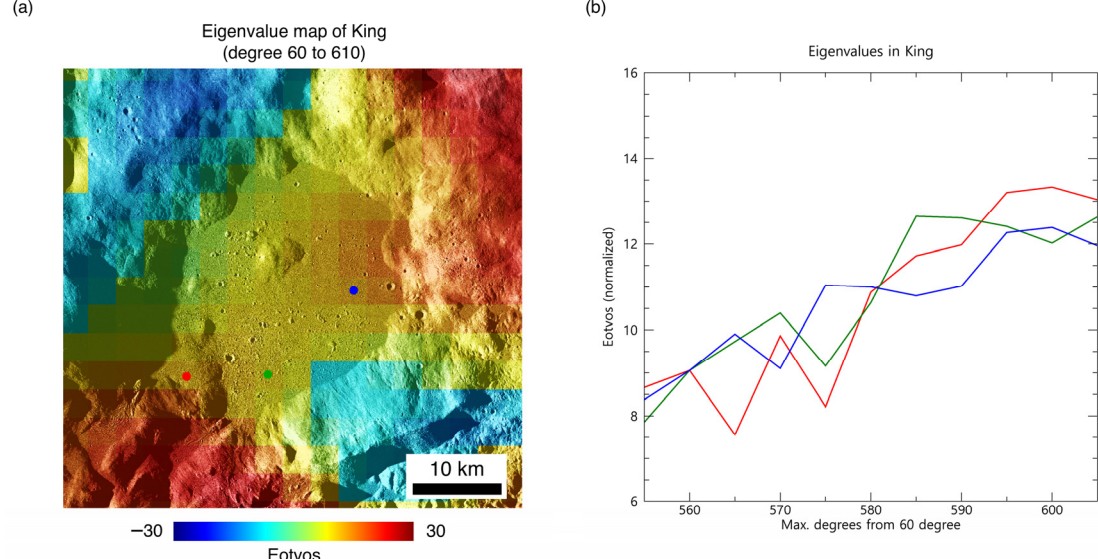

**Figure 6.** Eigenvalues map and eigenvalue variation of King pit region. (**a**) The King pond area is not a negative feature, but it cannot be said to be a strong positive feature. Only three points show mass deficit signals. (**b**) All three points are regions where pits are clustered, showing a positive trend. The graph is normalized at the 560th degrees to easily understand the trend change.

Unlike the other target areas, King Y has a small area of impact melt deposit and the largest number of impact melt pits per area [14]. Additionally, the distribution of fractures and the impact melt pits overlap [53]. Considering the results of gravity gradient analysis in this region and the distribution of a large number of impact melt pits in a small area, the possibility of a potential lunar cave network seems to be most likely.

The Stevinus crater has a diameter of 75 km. There was a mass deficit in only 4 out of 26 pits. Similar to Copernicus, most of the pits are clustered in the eastern part of the crater, but a positive trend was observed in the location unrelated to the cluster (Figure 7a,b). Additionally, unlike Copernicus and King, the three pits (red, green, and blue) are located hundreds of meters apart, but it is difficult to see these pits clustering.

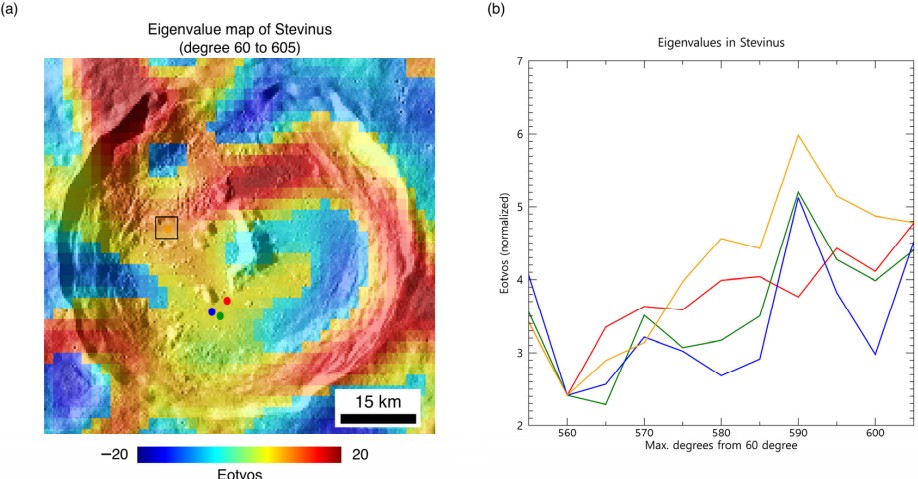

**Figure 7.** Eigenvalues map and eigenvalue variation of Stevinus pit region. (**a**) All four points indicating a mass deficit are located in the positive feature. The yellow point had low visibility, so a black box was additionally displayed. (**b**) Green and blue have weak oscillation, but an increasing trend appears in all four data points. The plot is normalized at 560 degrees for trend change visibility.

Tycho is located on the near side of the Moon and has a diameter of 80 km. Unlike the other three regions (Copernicus, King, and Stevinus), in the crater, all pits are distributed separately except for one region where five clusters are formed. As a result of the analysis, only 1 of the 24 pits showed a positive trend (Figure 8a,b).

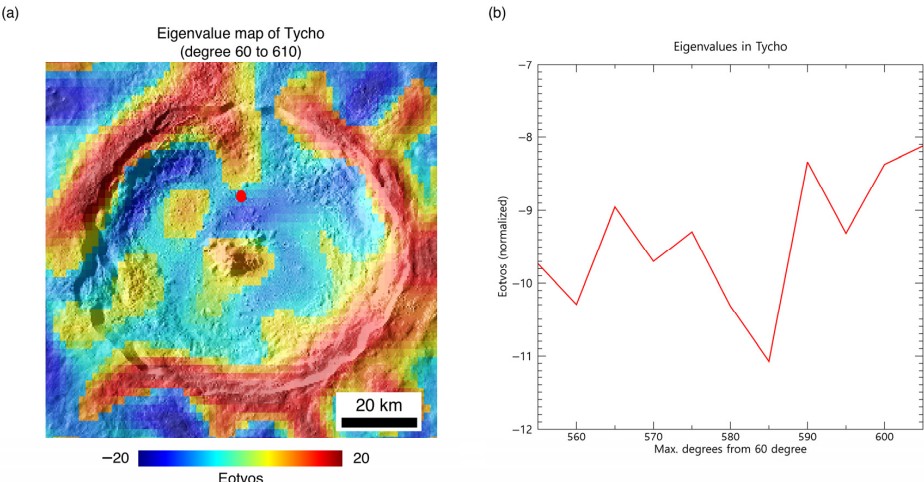

**Figure 8.** Eigenvalues map and eigenvalue variation of Tycho pit region. (**a**) Unlike other targets, a mass deficit appears only at one point and is located in a negative feature. (**b**) An oscillation pattern appears, but an increasing trend also appears.

## 4. Conclusions

In this study, we analyzed the eigenvalues of gravity gradient tensor changes for seven lunar craters to determine the possibility that an empty space, such as a lunar cave, may exist at shallow depths below the surface. We hypothesize the possibility of a network of small-scale caves, rather than large-scale ones. We selected an impact melt pit as a candidate and analyzed these pits to understand and determine gravity changes related to depth. The existing literature does not explore small-scale subsurface voids using GRAIL's gravity model; therefore, we devised a new method that could support such explorations. For the validation of the method, we compared our results to those of existing studies that analyzed known areas, such as the MHH.

From the analyses of the seven regions using our newly devised method, a mass deficit was detected beneath the pits in the four regions of the Copernicus, King, Stevinus, and Tycho craters. This study revealed the possibility of the existence of a cave network in shallow depth of the crust. Moreover, the possibility of the existence of a cave network is one of several reasons that can contribute to the mass deficit detected in the region.

We obtained meaningful results using the latest data on human space exploration conducted till date; however, our argument may seem weak due to the limitation of the absolute resolution of the images used in our study. However, because in situ exploration is essential for the reliable exploration of lunar caves, remote sensing can play a crucial role for similar future lunar explorations. In addition, when observing gravity and seismic waves in situ, more detailed internal information can be obtained than remote sensing, so it is necessary to actively consider it as a science payload for a landing mission.

**Author Contributions:** Conceptualization, I.-S.H. and Y.Y.; methodology, I.-S.H.; software, I.-S.H.; validation, I.-S.H.; formal analysis, I.-S.H.; investigation, I.-S.H.; resources, I.-S.H.; data curation, I.-S.H.; writing—original draft preparation, I.-S.H.; writing—review and editing, I.-S.H. and Y.Y.; visualization, I.-S.H.; supervision, Y.Y.; project administration, Y.Y.; funding acquisition, Y.Y. All authors have read and agreed to the published version of the manuscript.

**Funding:** This research was supported by the Basic Science Research Program through the National Research Foundation of Korea (NRF) funded by the Ministry of Education (NRF-2019R1I1A3A01063976). This work is also supported by a Chungnam National University research grant.

**Data Availability Statement:** The GRGM1200B RM1 gravity model is available at Geoscience Node of Planetary Data System (PDS). The MoonTopo2600p shape model is available at ZENODO [https://zenodo.org/record/3870924#.YaJL3MfP2F4]. LROC WAC images are available Cartography and Imaging Sciences Node of PDS.

**Conflicts of Interest:** The authors declare no conflict of interest.

## Appendix A

As mentioned in Section 3, It shows an example in which it is not easy to interpret the eigenvalues' increasing trend. There are cases where the oscillation is large, both an increasing and a decreasing pattern appear, and a case where there is little change in the trend. Additionally, in the case of the clone model, it is difficult to interpret because it shows a similar pattern to Figure A1.

We describe the analysis results of 5 clone models randomly selected out of 100 for each study area (Figure A2). In Clones 3 and 5 of Stevinus, green and blue are not clearly increasing. However, it can be argued that a positive trend appears for almost all clones.

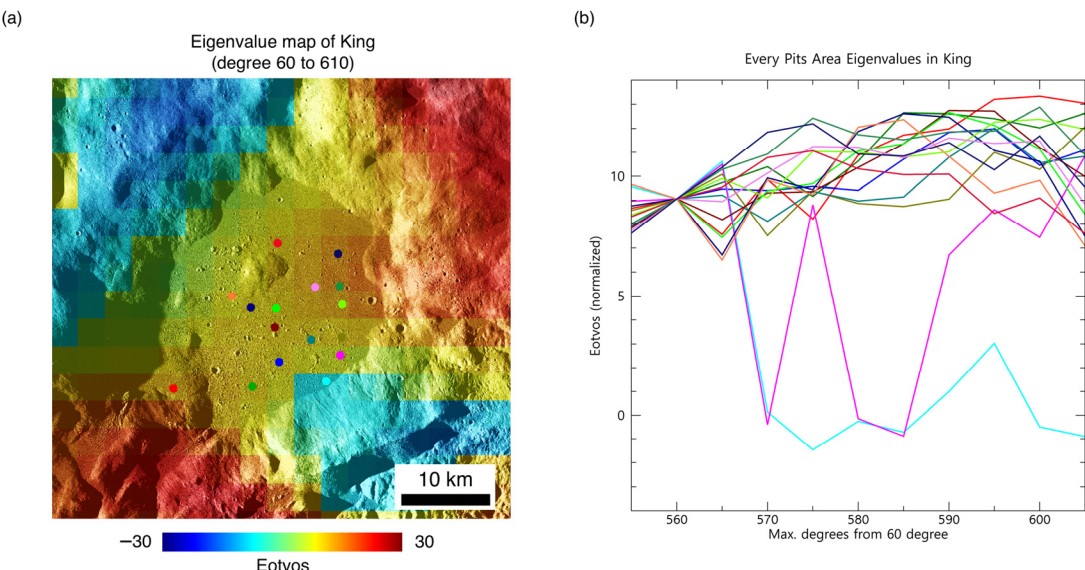

**Figure A1.** Eigenvalues map and eigenvalue variation of the King pit region. (**a**) In King, 62 pits form clusters in 16 locations. (**b**) Eigenvalue variation for 16 locations. The color of each line corresponds to the dot color in (**a**). It is normalized at 560 degrees, as shown in Figure 6.

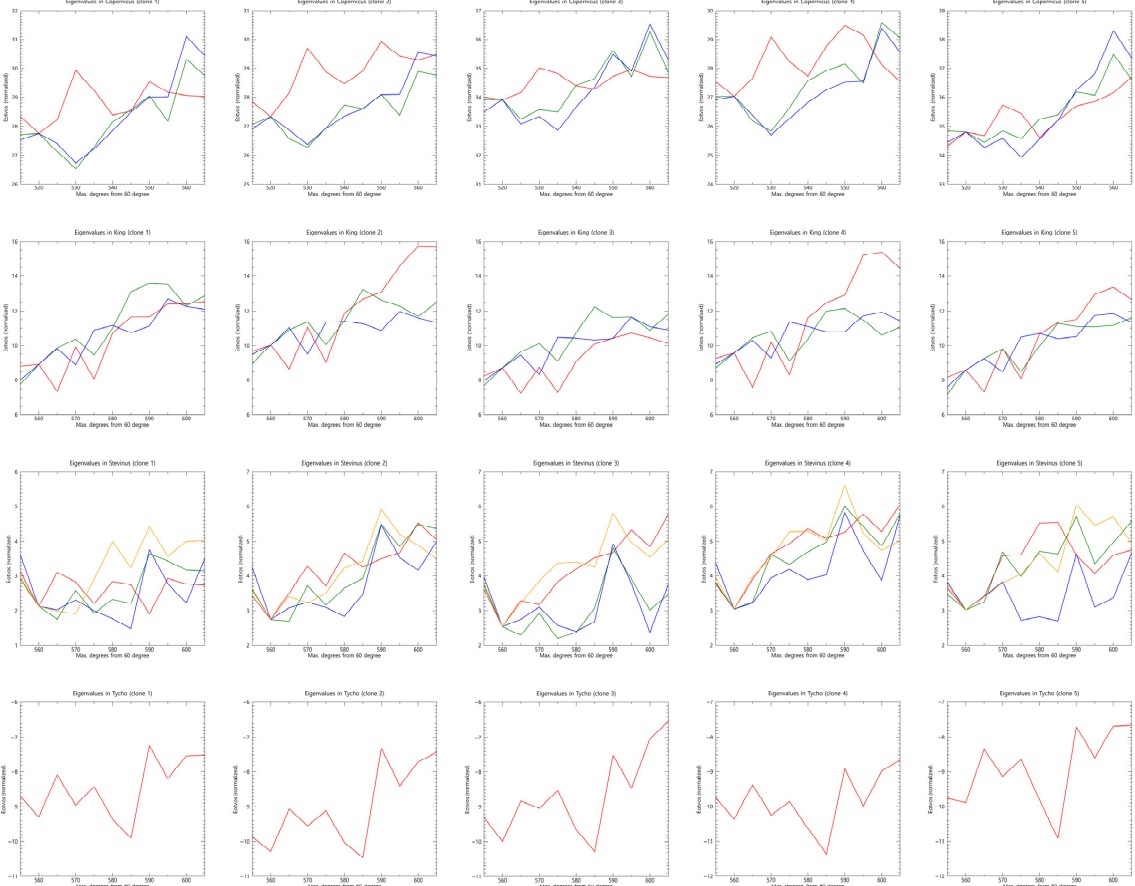

**Figure A2.** Clone model analysis results for the four regions in Section 3. A total of 5 were randomly selected from among 100 clone models. There are some points where the increasing trend is weak in Stevinus' clone 3 and clone 5, but it can be seen that a positive trend appears in almost all of the clone models analyzed. Each plot is normalized as shown in Figures 5–7.

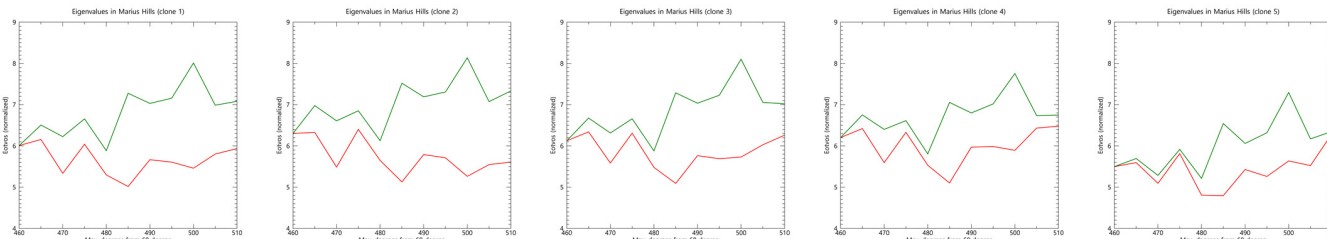

**Figure A3.** Clone model analysis of Marius Hills region. Five were randomly selected from among 100 clone models. The plot is normalized at order 460 for trend change visibility. The five clones of green have a similar pattern to Figure 4b. Red (MHH) can be said to be almost the same as Figure 4b, except for clone 2.

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
