# Peer review of "Plausible Detection of Feasible Cave Networks Beneath Impact Melt Pits on the Moon Using the Grail Mission"

_remotesensing, doi:10.3390/rs14163926_

Round 1

Reviewer 1 Report

I appreciate the time and effort that the authors have spent on their latest revision of their manuscript. I especially appreciate that they used my suggestion to look at clone models, despite the considerable computational efforts that this takes.

The manuscript has changed quite a lot between these two versions. The new manuscript is more focused, and reads really well. However, it is also my feeling that information has been lost. By showing only those results with clear positive trends, the manuscript is more compact, but I found the previous figures interesting in that they showed more trends for more pits. I understand that this is a choice of the authors.

I also must say that I still do not find the validation results of Marius Hills convincing. To me, there is no clear, unambiguous result that tells me that positive eigenvalue trends as presented by the authors can only be created by mass deficits with zero density, i.e. lava tubes. In absence of a pure synthetic test, did the authors consider to test eigenvalues in an area where there is no pit but that, in some way, looks similar to, for example, Marius Hills? What I am getting at it is some kind of validation of the eigenvalue trends: are we sure only and only see positive trends where there are lava tubes? Not when there is some other mass deficit? In the results section, there are plenty of examples with locations where there are pits but no positive trends; this is mentioned in the text (not shown anymore), but not commented upon. Not every pit has to be a (detectable) lava tube, but it does make one wonder.

While the results are tantalizing, I think this is a shortcoming in the story that the authors present. I don’t know what else to suggest than an unambiguous synthetic test. In that sense, the original Figure 3 in the first version of the manuscript was helpful (referring now to a comment in the reply letter). In my original review, I just wanted more background information on how to establish the trends, and if they really show lava tubes. The figure itself was informative, it just could use some improvements. I regret that the authors surmised I found it confusing. Again, while I am still not convinced that looking for trends in eigenvalues indicate lava tubes, the original Figure 3 went some way to at least convincing me there is something to looking at these eigenvalues over degree ranges.

I should have remarked on this in an earlier review, but for the admittances in Figure 2, did you downward or upward continue gravity to the average radius of your localized area? This may make a difference (although the values in Table 1 indicate mostly +/- 1 km, so it is probably OK). See McGovern et sl. 2002 and the correction that was needed in McGovern et al. 2004 (both JGR Planets papers, concerning Mars, but the concept is the same).

Line 258: change ”you can observed” to “you can observe”.

Line 270: I think it is the power spectrum of Bouguer that increases again after some degree. Here it is stated that the power of gravity models increases; but this will mostly be read as the power of the model that was used (akin to “free-air” power, and this does not increase except maybe at the end of the spectrum due to aliasing), not the Bouguer model.

Line 317: I understand the need to limit the errors of the gravity models. Chappaz et al. already were at the limit of the gravity models at the time. By limiting the degree range, can you be sure your resolution is not too coarse to detect changes of small features?

Line 352: can’t mantle uplift in the region be established by looking at crustal thickness?

The main manuscript could call out the clone results in more detail. The goal with such an analysis is to establish that errors in the gravity field models in the degree range considered do not contribute to variations in the eigenvalue trends: if the errors are small, all clones will be similar, and the trends should be close. The results in the appendix show that this is the case, so it is worth mentioning.

Reviewer 2 Report

The authors addressed my concerns and comments properly. I recommend its publication.

Author Response

We appreciate your comment on our paper. 

Reviewer 3 Report

The paper can be published.

Author Response

(The authors gave the same response as above.)

Round 2

Reviewer 1 Report

I again appreciate the changes that the authors have made to the manuscript following my comments. I am glad that they decided to bring back the former Figure 3. I think this helps in showing the reader that at least there is something to looking at the changes in eigenvalues. I also appreciate the additional textual changes to the text, where the authors are more careful about the interpretation of the findings.

I only have a few remaining textual comments. I would still also advocate for including results where the eigenvalue trends were less clear; at the least, include one or two plots to show that interpretation sometimes is difficult.

On line 325, I would use another word instead of stale. Maybe use “less certain”? Then again, that entire sentence is not really clear to me, and maybe the authors just want to leave it out.

Line 342 and beyond: “In this Section, we describe only four of the seven study regions (Copernicus, King,  Stevinus, and Tycho) where there is a clear mass deficit below the pit region, excluding those whose results are difficult to interpret or do not fit our hypothesis.”. You cannot say in a research paper that you exclude results that are difficult to interpret, or that do not fit your hypothesis. That sounds like you only look at those results that fit your hypothesis without considering those that would maybe falsity it! This is why I would advocate to either rephrase this part, or include results, as in a previous version, that were less clear (only some, not for all, though previously that was fine with me), to show the reader that interpretation is difficult.

Author Response

This manuscript is a resubmission of an earlier submission. The following is a list of the peer review reports and author responses from that submission.

Round 1

Reviewer 1 Report

The authors explore a way to detect possible caves on the Moon using the GRAVIL gravity model. In principle, this is in the right direction. However, as the authors stated in paper that the spherical harmonic degrees should be over 100,000 to detect a void of radius of 5 meters. The maximum degree of GRAIL is 1,200, which is far to meet the requirement. In the paper, the authors use the gravity gradients to amplify the short wavelength of the gravity field. In my opinion, if the signal is smoothed out in the gravity model, it is very difficult or even impossible to recover it. In short: I believe that the size of the detectable caves is limited by the degree and order of the gravity model. For instance, a spherical harmonic model to degree 1200 contains the gravity signal with wavelength longer than 3,000 m. Anomalous gravity signal of caves in size of a few to thousand meters would be not detectable.

Detecting mass anomalies below the Earth’s surface is a common practice. The Earth’s gravity models have been expanded 10,800 if the digital elevation model is combined. Even so, these models are still not very useful to determine density anomalies underneath the Earth’s surface because of its high error/signal at high frequencies. To improve the accuracy, the geophysical explorations have been using the airborne gravity and gravity gradient surveys, in combination with other data, such as the seismic wave data. In my opinion, in-situ gravity and gravity gradient observation may provide much more needed information for cave detection on the Moon.

Reviewer 2 Report

This manuscript presents a method and results to analyze gravity fields for the existence of lava tube networks on the Moon using GRAIL data. The manuscript carefully and very completely presents the evidence and implications of lava tubes. They then present a new method based on the analysis of gravity, using Bouguer and gravity gradient eigenvalues. They apply this method to a known area and then investigate several areas with likely lava tubes. For four of them, they find indications of the existence of such networks, while the results for the three others are less clear.

I find the method and results intriguing and as such they are worthy of publication in Remote Sensing. Before acceptance however, there are several issues that I would like the authors to address. This pertains to the method, which is not entirely clear to me, and to the robustness of their results.

While I think I generally understand the method, more details are needed. The authors need to clearly define how they obtain their eigenvalues. Is this similar to the Andrews-Hanna papers, where the horizontal gradients are used to determine eigenvalues? Or is this something different?

I would also like more explanations on figure 3. Again, I understand the reasoning and conclusions, but it is not entirely clear how this example was set up. Synthetically? What is the range in y-values for Figure 3b? When is a trend significant enough? In addition to this, I wonder if the authors can apply their method to an area on Earth. The EGM model series are highly accurate and at great resolution. Moreover, we have access to known lava tube networks. If the authors could use such an example, it would be a great help to establish that their method works. It is not entirely clear to me for example how exactly the Marius Hills example matches the earlier results from GRAIL (which already were at the edge of resolution).

As for the results of the areas, I find the figures in the appendix to be clearer than the eigenvalue maps. As a reader, it is difficult to discern trends from the maps with different degree expansions, whereas the appendix figures at least have trends. But, it is not clear how these figures in the appendix were constructed. Does “mean” indicate the average of the maps, or taken from individual pixels (such as, the coordinates indicated in Table 1)? From these figures, some trends are not so very clear (even if they are suggestive; lots of ups and downs are difficult to interpret and cannot readily be density highs and lows in shallower and shallower layers). And the authors talk about trends in parts of the maps in the main manuscript, which are not clear to the reader. Rather than showing the full maps, would it be an idea to show changes in eigenvalues in maps? So, one would have 4 full maps (say, in degree ranges, 60-575, 60-580, 60-585, and 60-590, but differences showing 575-580, 580-585, and 585-590. Would that indicate regional trends clearer?

Concerning the robustness of the results, the authors already indicate themselves that several of the areas under investigation show the sought-after positive trends, but they cannot be sure they are due to lava tube networks. The figures in the appendix show standard errors, but first of all, it is not clear how these are obtained, and second of all, not much is done with them. They mostly seem to be a straight offset from the central line in bold anyway. And this does not really address the robustness. In the Chappaz et al. paper, they used clones of the gravity fields, which are equivalent expressions of the gravity field taking errors into account. In other words, they are from covariance propagation and can be used to probe variations in gravity due to estimated errors in the coefficients. For the new gravity field model used here, such clones are also available, and I think it would really help this study if the authors repeat the trend analysis using these clones. For each area, they can compute a trend for each clone, and then see if the trends hold up: are they statistically significant? This may even give more information about the areas they now have to exclude.

In addition, how big is the influence of fixed steps and degree cut offs? In other words, if we see a trend for ranges 575-580, 580-585, 585-590, do we still see the same trend if these degrees shift by 1 or 2? Or, in yet other words, are trends between only 5 degrees in spherical harmonics meaningful? It would be good to establish this with the clones, or maybe even a synthetic example.

I have additional comments that I will list here below, indicating line numbers in the manuscript.

Line 11: use “expressed in spherical harmonics”.

Line 12: The use of “Therefore” implies something following earlier reasoning, but the hypothesis of networks does not follow from the previous sentence which is about spherical harmonics and scales.

Line 15: the use of “weak” does not put confidence in the results!

Lines 66-67: use “insides” and change to plural.

Line 100: I suggest to use “with a void height of”.

Line 128: In addition to my comment about clones, one could additionally compare the trends with the 1500E model to establish whether they are robust. In general though, the models are very similar up to the degrees used (listed in Table 3) so the clone approach is probably the better to probe variations.

Line 138: Maybe the authors could repeat the equations from Lowrie used? To make clear how different variables act. Note that a density contrast of 2,550 kg/m^3 (zero, and average crustal density) is a lot!

Figure 1: the authors acknowledge that these are resolutions GRAIL cannot get at, hence the idea to think of networks. This seems more a necessity from resolution then something that follows from the analysis. And it is not clear what kind of signal a network would give (other than, a bigger signal, probably). Also, it is not entirely clear what g0 is: isn’t that just the point directly above the mass deficit?

Line 173: add “that” before “form”.

Line 216: the authors could include the equation here for admittance. Also, how exactly are the densities obtained? From the effective density spectrum, such as used by Wieczorek et al. in their GRAIL crustal paper? Or with a different method? Expressing gravity from topography and minimizing the trend in the area by choosing a density? More details are needed here.

Figure 2: The trend for Crookes really stands out, as the authors indicate. But what about density variations with depth such as discussed in Besserer et al. (2014)? They found these kind of admittance curves to indicate density varying with depth.

Line 217: note that the Simons et al. 1997 paper, reference [45], uses a very different method than the Simons, Wieczorek, and Dahlen papers.

Line 240: add a reference to the value of degree 60 commonly used (like an Andrews-Hanna paper, but low or high pass filters are used for different purposes).

Line 265: it can indeed be noted that lunar power spectra start to increase after certain degrees. This is however not necessarily related to the error increasing, but it is more related to the reference radius (see recent papers by Sprlak et al.). Still, I think it is OK to use this. The authors could also use the “degree strength” reported in the gravity field model papers, where degree strength indicates the maximum spherical harmonic degree to be used locally, from considering signal and local error.

Line 282: I find the descriptions of the degree ranges in gradients not clear. Eventually, I realized that a range of 20 degrees is used, from the maximum degree in Table 3. It is just better to explicitly state which degree ranges were used, so for example, for King that would perhaps be 585-605 (the max degree is 606 but if the step size is 5, then 605 is an easier number, I assume).

Figure 4 and the other map figures: as stated above, the eigenvalue maps are not clear to me to figure out trends. Also, it is very difficult to see the pink crosses: I cannot see them in a printed version, and had difficulty with the PDF. Some of the features described in the text are also not clear to the reader who is not intimately familiar with the area under consideration.

Figure 7: a positive trend is listed in the caption, and to me this is an example of trends that are not clear to me from the maps.

Line 444: with the negative trend appearing as degree and order increase, does that mean that the lava network would be deeper? Is that reasonable, because the resolutions/depth differences would be on the order of km maybe? This also relates to my earlier comment about if the trends are still the same if the used degree ranges shift by a few degrees. A degree and order 600 model has a resolution (depth) of 9 km (180/600 * 30 km, with 30 km from 1 degree at the equator), and a degree 605 model results in 8.92 km, only 0.1 km difference. Do the models have such “differential resolution”?

Line 501: there is an extra period between “peak” and “portrayed”.

Appendix figures:

A1: the negative trend in Crookes is not so clear as it is very weak. I am also not so sure that the full range eigenvalues (the right plots) add much. They are not really discussed (they do give context, of course).
In the captions, the use of “50-lower range” is not so clear, and I suggest to just explicitly state the range used.

Reviewer 3 Report

The paper presented is well written and can be characterized as having high interest to the readers. I have some comments and questions to the authors.

1) The authors write that " shallow depth information can be effectively expressed using the GRAIL model". It is well known, however, that there exist a continuous variety of sources equivalent with respect to the external gravity field (the so called Novikov's  lemma).  The different  deeply localized distributions of masses can generate the same gravity (and other potential fields) in the upper subspace.   

2) What is the range of high degrees and orders which can ensure an adequate lunar cave structure? In my opinion, recovering the lunar structure (and cave structure first of all)  relates to ill-posed  problems and needs a regularization algorithm.

3) The formula for admittance sis recommended to be given in this paper. 

4) I think that the implementation of the multitaper method in this work should be  justified. The gravity and topographic data can be interpreted directly without analyzing their spectra because of  many arbitrary errors in Fourier coefficients.

5) The gravity gradient tensor analysis can be accomplished, in my opinion, if and only if  the  gravity and topographic data are obtained with very high accuracy. Computing the derivatives of an approximately given function, as it is well known, relates to ill-posed problems. It means that the tensor components should be computed with the help of a regularization procedure.  The GRAIL model is an effective tool for gravity analysis and synthesis but the reconstruction of the shallow distributions is a very complicated process and it requires elaboration of some new effective tools for finding stable approximate solution to inverse problems of geophysics.

5) Why is the maximum degree and order equal to 507? The authors used the model with SH up to 1200  harmonics.  I recommend the authors to give a more clarified explanation to this fact.

6) The number of images on Figures 5-11 should be reduced.

Round 2

Reviewer 1 Report

The revised paper is improved, but I’m still not convinced that we can detect the underground holes (let’s say 5m to 50 meters diagram) using the spherical harmonics to degree and order of 500-600, which has a wavelength of 5 km. The wavelength of the model is 100 to 1000 times longer than the wavelength of gravity signal the holes produce. In theory, the voids produce gravity anomalies at all wavelengths. Unfortunately, the gravity signals are averaged out in the spherical harmonic modeling. To catch such details of the gravity field, the spherical harmonic model should go to much higher degree and order. A low orbit gravity gradient mission may be helpful.  

If the paper is to be published, the authors should answer the questions how to reveal the gravity information from a model who’s wavelengths are 100 to 1000 time longer than the signal sought for.

Reviewer 2 Report

The authors have made several revisions to the manuscript that have improved it in my opinion. The figures for the different areas under consideration are much easier to understand now. The explanations about obtaining the local densities, with added formulas, also helps to understand how the values in Table 2 were obtained. But despite this, I still have reservations about the presented method and results. In my original review, I made several suggestions to try and show that the results are robust. The authors indicate they will leave this for future analysis due to the time it takes to do the analysis. I do not find this a sufficient reply. In the case of looking at clone models, the authors would need to rerun their analysis with the only difference being a set of statistically slightly different models. It may be that this will require some time, but I do not know how much. Nor does it seem to be an onerous task. I think this would go a long way to showing that the results are robust. If the authors have different suggestions, that would be fine too. But a clear validation is currently lacking. The case for Marius Hills that the authors rely upon is still not convincing to me. And an even more basic case, where they show that they can capture a lava tube network (from synthetic modeling, or otherwise), would be even better. It is now hypothesized, almost, that the trend in eigenvalues of the gradient can be associated with underground networks, but it is not clear that that is the only association. At least the previous version had Figure 3 which showed this; this should at least be brought back into the manuscript. But there should be a clearer indication for the reader that indeed a positive trend in eigenvalues can be associated with voids (and then it may still need to be inferred that in certain locations, those voids could be lava tube networks).

There is also no clear quantitative way of deciding which areas show trends in the eigenvalues. I should have maybe remarked on this in the review of the original version, but the presentation of the areas (which, I agree, is much better) now shows the eigenvalue map and trends that are used for the conclusions. A quantitative indication of trends (again, clone models, or some other method, would help a lot) would also make their conclusions more robust.

I really think these aspects should be considered before publication.

In the following, I will list comments and suggestions (some of which may be a repeat of my summary above), with line numbers indicated.

In the abstract, lines 12 and 13 are contradictory. It is stated that in GRAIL models it is difficult to detect empty spaces, but this is followed by the statement that it is proposed that a network of lava tubes exists.

Line 157: the use of “Therefore” is strange here. It is stated that a small void would need a high SH degree. But then it does not necessarily follow that they must form networks. The two are not related. One could however state that only networks give a strong enough signal to be seen in the current gravity field models. Line 161 hints at this, but instead of “expressed” I would use “detected in current gravity field models”.

Line 165 explains delta g0 and it is stated it is the surface above the void position, but shouldn’t it be vertical gravity?

Line 209: the wavenumber cannot readily be replaced by the spherical harmonic degree. The areas here may be small enough that a Cartesian approach works, but the authors should explicitly state this.

Line 229: It is not clear to me how eq. (3) is used. Directly? Or is it localized in the same way as the localized admittance from the gravity and topography models were computed? And the connection from wavenumber k to SH degree is not clear, especially for cap sizes of 10 degrees which are considerable and for which curvature would matter. Wieczorek and Phillips 1998 indicate how to compute uncompensated topography in SH. Wouldn’t this be better to use here? The densities in Table 2 do seems reasonable, but more context is required.

Line 282: I do not quite understand this sentence. What does “To ignore the error” mean? Which error do the authors refer to here? Do they mean, to minimize the error made in the approximations?

Line 293: I find the statements here confusing. The green and red dots refer to two pits, right? But in this line, green is indicated as being another point. Does that mean it is a different point for comparison where there is not a pit? But if that is the case and there is a positive trend for the green line, then doesn’t this mean that a positive trend is not exclusively related to the existence of pits? I still do not readily see how this image indicates that the gradient method is consistent with the results of Chappaz et al. The red line in Figure 3b only has a weak positive trend after maximum SH degree 484 or so. To me, this does not look like very strong evidence.

I have suggested several ways of trying to establish the validity of the method. I understand that doing an analysis with Earth models may be difficult (but it is difficult for me to understand how atmospheric signals could affect the high degree of Earth gravity field models. Would that not affect low-degree time-varying signals more?). But a synthetic test of some kind should be doable, where one has full control over the signals. Or, as indicated, a clone analysis using existing clones for the gravity field model, as that would increase confidence in the reported signals. I also understand that the computations can be time consuming, but this is not a valid reason not to perform them, since obviously the original analysis has already been undertaken (so the codes etc. are there, they would need to be run with a set of different, closely related models).

Line 312: I don’t think this has truly been shown, that at the selected maximum degrees the errors remain small. It is not clear that the trend is statistically significant. It should be straightforward to show if this is the case with error propagation or clones, or some other way that the authors may prefer.

Figure 4: some of the pits are clustered closely together. It can be questioned whether the gravity field models have enough resolution to have distinct pixel values for the eigenvalues at each location. The authors should comment on this. For example, are the negative trends in Figure 4b statistically independent? Or are the locations so close together that they are all correlated? This of course is the case for all the areas, not just for Figure 4.

In Figure 4, the dotted lines are standard deviations, but it is not clear how the upper and lower bounds are computed. They are also not used at all in the manuscript, it seems.

For Crookes it is stated there are no clear trends. But some curves do seem to have a positive trend, even if weakly. And I do not see how this is different from Figure 3b, where it was stated that for Marius Hills, the earlier results from Chappaz et al. were confirmed. There is no clear quantitative criterium for deciding the trends, which makes it subjective to judge if there is one or not. This of course goes for all the other areas as well.